# Comparison of Plate Waste between Vegetarian and Meat-Containing Meals in a Hospital Setting: Environmental and Nutritional Considerations

**DOI:** 10.3390/nu14061174

**Published:** 2022-03-11

**Authors:** Andrew Berardy, Brianna Egan, Natasha Birchfield, Joan Sabaté, Heidi Lynch

**Affiliations:** 1Center for Nutrition, Lifestyle and Disease Prevention, School of Public Health, Loma Linda University, Loma Linda, CA 92350, USA; began@students.llu.edu (B.E.); jsabate@llu.edu (J.S.); 2Roche Diagnostic Solutions, Clinical Development Core, Clinical Development Medical Affairs, Tucson, AZ 85755, USA; birchfin@roche.com; 3Department of Kinesiology, College of Health Sciences, Point Loma Nazarene University, San Diego, CA 92106, USA; hlynch@pointloma.edu

**Keywords:** food waste, global warming, vegetarian meals, hospital setting, plant based, sustainability, public health

## Abstract

Vegetarian diets can satisfy nutritional requirements and have lower environmental impacts than those containing meat. However, fruits and vegetables are wasted at higher rates than meat. Reducing both food waste (FW) and the environmental impacts associated with food production is an important sustainability goal. Therefore, the aim of this study was to examine potential tradeoffs between vegetarian meals’ lower impacts but potentially higher FW compared to meat-containing meals. To examine this, seven consecutive days of plate FW data from Loma Linda University Medical Center (LLUMC) patients were collected and recorded from 471 meals. Mean total FW and associated greenhouse gas emissions (GHGE) were higher among meat-containing meals (293 g/plate, 604 g CO_2_-eq/plate) than vegetarian meals (259 g/plate, 357 g CO_2_-eq/plate) by 34 g (*p* = 0.05) and 240 g CO_2_-eq (*p* < 0.001), respectively. Statistically significant differences were observed in both FW and associated GHGE across major food categories, except fruit, when comparing vegetarian and meat-containing meals. Overall, vegetarian meals were preferable to meat-containing meals served at LLUMC both in terms of minimizing FW and lowering environmental impacts. Other institutions serving vegetarian meal options could expect similar advantages, especially in reduced GHGE due to the high CO_2_ embodied in meat.

## 1. Introduction

Human activities cause global environmental changes that threaten to disrupt the stability of the Earth’s systems, leading to potentially disastrous consequences [1]. This recognition has prompted a widespread call for emergency action to limit global temperature increases, restore biodiversity, and protect health [2]. Food systems are responsible for between 19 and 37% of global anthropogenic greenhouse gas emissions (GHGE), depending on what is included in the estimate [3,4]. A recent estimate attributed 34% of global GHGE to food systems, with 71% coming from agriculture and land use, and the rest from downstream supply chain activities [5].

Yet, current practices of food production and distribution are insufficient, as there are 815 million people globally, or one in nine, who are undernourished [6]. In order to end hunger, different scenarios predict that between 3 and 20% more food production will be necessary, depending upon the approach, increasing the associated environmental impacts [7]. This challenge will only become more difficult as the global population continues to expand to approximately 9 billion people [8].

Reducing the consumption of animal-based foods is a possible measure to reduce environmental impacts while improving health outcomes, with the potential to reduce diet-related GHGE by between 33 and 51% in the United States [9,10]. A systematic review found that vegan diets could reduce GHGE by up to 70%, land use by up to 86%, and water use by up to 70% [11]. Another review found that along with improved health, shifting from current omnivorous dietary patterns to vegetarian or vegan diets increases environmental sustainability while also improving health [12].

The Academy of Nutrition and Dietetics considers appropriately planned vegetarian diets to be healthful and nutritionally adequate for all stages of the life cycle [13]. Consuming vegetarian or vegan diets has been shown to lower risk for developing obesity [14], cardiovascular diseases [15], hypertension [16], type 2 diabetes [17], and metabolic syndrome [18]. These health-protective effects may be due to the higher nutrient quality typical of plant-based diets [19]. Notably, vegetarian and vegan diets tend to be lower in total fat, saturated fat, monounsaturated fat, dietary cholesterol, protein, alcohol, and sodium, and higher in polyunsaturated fat, fiber, and iron [19]. This is likely because plant-based diets tend to be higher in fruits, greens, and pulses; subcategories of vegetables [19]. In addition, plant-based diets have been found to sufficiently support athletic performance while also contributing to better overall health and reducing environmental impacts [20,21,22].

Nonetheless, certain nutrients are less bioavailable or less frequently consumed on a vegetarian or vegan diet. For example, as non-heme iron (found in plants) is less bioavailable compared to heme iron (from animals), the Recommended Dietary Allowance for iron for vegetarians and vegans is 1.8-fold greater than that for omnivores [23]. Additionally, vegans (who exclude all animal products) must be mindful to consume foods fortified with vitamin B12 or take a vitamin B12 supplement as this vitamin is not present in plant foods [13]. Lacto-ovo vegetarians typically consume at least the recommended intake for calcium, while vegans may risk insufficiency. Furthermore, vitamin D is not abundant in food and is a nutrient for which the use of supplements is frequently advised, regardless of dietary pattern [24].

Higher diet quality, as measured by the Healthy Eating Index, is associated with higher food waste (FW), primarily in the form of fruits and vegetables [25]. FW is a significant challenge, as 32% of all food produced in the world by weight or 24% by kilocalories (kcal) is wasted [26]. If global FW were treated as its own country, it would be the third largest emitter of GHGE, behind China and the United States, occupy 30% of the world’s agricultural land area, and use the equivalent water of the annual discharge of the Volga river in Russia (i.e., 250 km^3^) [27]. These FW statistics represent wasted resources and wasted opportunities to eat health-promoting foods, which comprise a large portion of total waste. In fact, the average global FW per capita per year could fulfill a person’s dietary recommended intake (DRI) of 25 nutrients for 18 days [28]. Based on the types of food wasted, that amount of FW contains between 25 and 50% of the DRI for vitamin C, K, zinc, copper, manganese, and selenium for a person [28].

It is important to understand possible tradeoffs when promoting a solution to one problem to ensure it does not exacerbate another. For example, given the relatively high proportion of fruit and vegetable waste compared to meat waste, and the small proportion of vegetarians in the general population, could there be higher FW as a result of reducing meat-containing meals? Moreover, would the environmental impacts associated with that FW be substantial enough to negate the benefits of serving vegetarian meals as the default in large institutional settings? Although there are publications that assess hospital FW, its environmental impacts, and techniques for FW reduction, no literature has previously examined these questions [29,30,31,32].

Loma Linda University Medical Center (LLUMC) provides a unique setting to examine the potential tradeoffs associated with serving lower environmental impact foods with potentially higher FW compared to higher environmental impact foods with lower FW. Unlike many hospitals, LLUMC serves lacto-ovo vegetarian meals to patients by default for the first 24 h upon admission. However, patients have the option to reject the default and can choose their preferred meal items from standard menus, which include meat, after 24 h. As such, the aim of this case study was to examine the differences in FW and GHGE between vegetarian meals and meat-containing meals served in a hospital setting.

## 2. Materials and Methods

A plate waste audit was performed by Loma Linda University dietetics graduate students across seven consecutive days, from September 6 to 12, 2020 at LLUMC. Plates audited included those served at breakfast, lunch, and dinner and were provided by meal services on three hospital floors, which housed patients with the fewest special dietary orders (e.g., liquid diets or “nil per os” (NPO, nothing by mouth)). At least 20 plates were audited, upon tray return and prior to disposal, after each meal service. Each tray was assigned a de-identifiable number and meal type (i.e., “meat-containing” or “vegetarian”) based on the food items listed on the tray ticket, which reflected the patient’s menu order. Trays returned without tray tickets and no remaining meat items were categorized as unknown meal type. Floor number and diet order (regular or therapeutic) were also noted for each tray. Institutional review board approval was not needed since no patient-identifying information was collected. For each tray, all remaining individual food items were removed and individually weighed in grams before being discarded. Liquid diet trays were excluded from measurement due to the high proportion of total weight from liquid.

Data processing included removing container weight values from FW measured in containers. LLUMC also provided recipes for composite foods such as cooked entrees and soups, which were used to determine the proportional weights for individual ingredients (e.g., spinach, cheese, and egg white for spinach quiche). In addition to FW weight, GHGE were estimated using a combination of SimaPro life cycle assessment software and published literature used to fill any gaps where SimaPro did not have appropriate data [33,34,35,36,37]. The life cycle assessment (LCA) studies used for GHGE estimates all had cradle to farm or manufacturer gate system boundaries and reported results using a weight-based functional unit. The parameters of LCA studies included were system boundaries from cradle to farm or manufacturer gate or distributor, excluding retail, consumption, and disposal, and used a weight-based functional unit and attributional assumptions.

Descriptive and statistical analyses were conducted using IBM SPSS Statistics for Windows, version 28 (IBM Corp., Armonk, NY, USA). Tests for assumptions of normality and homogeneity of variance were performed using Kolmogorov–Smirnov and Levene’s test, respectively. To examine between-group differences in total FW and total GHGE, independent *t*-tests were performed. Values that were ±2.5 standard deviations from the mean were considered outliers. Visual assessment using boxplots indicated that there were no outliers, defined as values that were ±2.5 standard deviations from the mean. Post hoc exploratory analyses were also conducted using independent t-tests to compare between-group differences for FW and GHGE by primary food categories. The exploratory analyses were considered secondary analyses, which were not driven by hypothesis testing; therefore, the significance level was not adjusted for multiple comparisons. In addition, effect size was calculated using Hedges’ g for all primary and secondary outcomes. Data are reported as the mean ± standard deviation and the level of statistical significance was set at *p* = 0.05.

## 3. Results

Plate data were analyzed for 447 patient trays of the 471 that were collected. Twenty-four patient trays were excluded from analysis due to unknown meal type and absence of identifying characteristics (e.g., leftover meat or a tray ticket). Key findings of this study were that mean total plate waste was higher among meat-containing meals, and that the associated GHGE was lower among vegetarian meals. 

### 3.1. Food Waste

#### 3.1.1. Descriptive Statistics

The corresponding means and standard error of mean are presented graphically in Figure 1. The data for total FW were not normally distributed for either group (*p* < 0.05). Skewness of variables prevented transformation to a normal distribution. Non-parametric tests, such as the Mann–Whitney U test, resulted in unacceptable values (U > 10,000) and are most appropriate for analyzing ordinal data. Thus, non-parametric testing was excluded from the analytical approach. However, based on the central limit theorem, with adequate sample sizes (*n* ≥ 30), violation of the normality assumption is unlikely to affect statistical findings. Therefore, parametric tests were acceptable due to the large sample size. Homogeneity of variances was observed (*p* = 0.64). Descriptive statistics for FW (g) by each food category and meal type are provided in Table A1 in Appendix A.

Total mean FW was greater among meat-containing meals (292.51 ± 180.77 g/plate) compared to vegetarian meals (258.46 ± 186.09 g/plate), with a mean difference of 34.05 g/plate, t(445) = 1.96, *p* = 0.05, g = 0.19 (Figure 2). The largest FW source for meat-containing meals was vegetables and fruit, while vegetarian meals had the most FW from grains and vegetables. 

#### 3.1.2. Exploratory Analyses

Exploratory analyses revealed significant differences in FW and GHGE between groups for analyzed food categories except fruit (Table 1). Vegetable and dessert waste were significantly greater among the meat-containing meals, while grains, dairy, egg, and plant protein waste were significantly greater among the vegetarian meals.

There were statistically significant differences between meat-containing meals and vegetarian meals for every major food category shared by both meal types except fruit.

### 3.2. Global Warming Potential

Descriptive statistics for GHGE by food category and meal type are provided in Table A2. The difference in total GHGE was also compared between meal types. The data were not normally distributed (*p* < 0.001) and homogeneity of variance was not observed (*p* < 0.001). The ratio of the meat-containing meals to the vegetarian meals is 1.1; thus, this violation is unlikely to affect statistical findings. Total GHGE was significantly greater for meat-containing meals (604.20 ± 643.45 g CO_2_ eq) compared to vegetarian meals (356.66 ± 376.98 g CO_2_ eq), t(445) = 4.995, *p* < 0.001, g = 0.47 (Figure 3).

Total GHGE were significantly higher for FW from meat-containing meals than for vegetarian. The highest contributor to GHGE was animal protein, followed by dessert. The highest contributor to FW from vegetarian meals was dairy, followed by dessert.

GHGE from both meat-containing and vegetarian meals’ waste had a high standard error of means. GHGE from vegetarian meals’ waste was much lower than that from meat-containing meals’ waste.

GHGE was significantly greater among meat-containing meals for the vegetable and dessert food categories compared to vegetarian meals. GHGE was significantly greater among vegetarian meals for grains, dairy, egg, and plant protein.

GHGE associated with plate waste showed statistically significant differences across all food categories except fruit when comparing plate waste from meat-containing meals to plate waste from vegetarian meals.

## 4. Discussion

The objective of this study was to examine the differences in FW and GHGE between vegetarian meals and meat-containing meals to determine if greater FW among vegetarian meals offset the associated environmental benefits when compared to meat-containing meals. Analysis of plate FW failed to demonstrate evidence that vegetarian meals are associated with more FW or corresponding GHGE. Therefore, there does not appear to be a tradeoff or downside to providing vegetarian meals to patients by default for the first 24 h following their admission to a hospital setting from this perspective. 

Previous work has not investigated the possibility that extra FW would be generated by providing vegetarian meals by default, which could potentially negate the environmental benefit of doing so, when compared to serving meat-containing meals by default. Only a couple of studies have reported actual FW in hospitals at the item level [29,32]. Change in meal service style from traditional foodservice to room service can reduce FW by approximately one-third [30]. GHGE from meals in a hospital setting were estimated to be approximately 5 kg CO_2_-eq per day for a 2000 kcal diet, with a range between approximately 0.5 and 8 kg CO_2_-eq for liquid diets and high protein diets, respectively [31]. GHGE from plate waste itself amounted to an average of approximately 1 kg waste per patient per day, which was associated with approximately 1.8 kg CO_2_-eq [32]. Plate waste refers to food that was served to a patient but not consumed, as opposed to tray waste, which includes other non-food waste, such as packaging [38]. Numerous studies indicate that the GHGE from animal-based foods are higher than those from plant-based foods [9,10,33,35].

However, FW from vegetarian meals in this study was approximately 11% lower than that from meat-containing meals, which represents a difference that is approximately half the reduction in FW observed in another study that examined FW reduction from a transition to room service rather than traditional foodservice [30]. Additionally, the average GHGE from daily plate waste per patient reported here for meat-containing and vegetarian meals was approximately 36% and 21%, respectively, of the average GHGE per day for a 2000 kcal diet in a hospital setting reported in another study [31]. In addition, the GHGE per patient per day in this study of approximately 1.8 kg CO_2_ eq for meat-containing meals matches the value reported in another study of hospital FW and emissions of 1.8 kg CO_2_ eq per patient per day [32].

Meal provision is considered an “environmental hot spot” in hospitals [39]. To address this, it has been proposed to list vegetarian meal choices first on menus and to offer more vegetarian meal options in hospitals [39]. The European Society for Clinical Nutrition and Metabolism (ESPEN) affirms the importance of providing vegetarian meals and other specialized dietary patterns to be respective of religious and dietary preferences to patients as well, noting the increased demand for vegetarian meals by patients [40]. Providing vegetarian meals in hospital settings may have synergistic benefits beyond reducing FW and environmental impacts by also promoting health.

California licensed health care facilities and state prisons are required by law to make available “wholesome, plant-based meal options” to meet patient needs and follow physicians’ diet orders according to CA Senate Bill No. 1138 [41]. Additional California law (Senate Bill No. 1383) sets targets for statewide organics recycling to reduce short-lived climate pollutants, such as methane from food waste sent to landfill [42]. The American Medical Association passed a resolution in 2017 (H-150.949) calling on US hospitals to “improve the health of patients, staff, and visitors by providing a variety of healthy food, including plant-based meals” [43]. As hospitals work to comply with such laws and resolutions, this study demonstrates that serving plant-based or vegetarian meals may provide overall reductions in FW and GHGE generated from meal service.

United States federal regulations require that hospitals provide “a nourishing, palatable, well-balanced diet that meets the daily nutritional and special dietary needs” of patients (42 Code of Federal Regulations 483.35), informed by the recommendations of a qualified registered dietitian, and that menus meet nutritional needs as recommended by the Food and Nutrition Board of the National Research Council, National Academy of Sciences [44]. While maintaining compliance with such regulations, as well as specific state regulations, there may be particular advantages conveyed by providing vegetarian meals. For example, there is a clear connection between proper nutrition and a healthy immune system to protect against infections [45]. Of particular relevance currently, healthy diets as measured by the Plant-Based Diet Score are associated with lower risk and severity of COVID-19 [46]. Health care workers (who often eat meals provided by the hospital cafeteria) who reported following plant-based diets and low-meat diets also had lower odds of moderate to severe COVID-19 [47]. 

There were some limitations to this study. Some food categories were excluded from exploratory statistical analysis due to inherent differences between meal types (e.g., vegetarian meals contained no animal protein). Some additional food categories were excluded due to having near negligible mean values. The food categories excluded were meat analogues, animal protein, sugars, condiments, and sauces. The larger amount of plant protein waste from vegetarian meals was expected, as these trays were more likely to contain higher amounts of plant proteins including peanut butter, tofu, black beans, brown lentils, and hummus. 

Future research should include measurements of initial food weights to understand the proportion of each meal wasted and facilitate comparison across meals with different starting weights. It may also be useful to explore differences when correcting for kcal content of meals. Additional research could also examine correlations between meal type (e.g., liquid, dysphagia, cardiac, and low sodium), patient ward (e.g., surgery and intensive care), and outcomes (e.g., length of stay), as well as explore differences based on demographic factors such as sex or age.

Generalizability of the findings from this research is likely most applicable to other hospitals and similar settings where food is provided, but from fairly limited options and with few if any alternatives. In a hospital setting, there are often limited choices and the consumer may be feeling unwell, both of which increase the likelihood of them wasting food. In contrast, consumers are normally able to choose from a wide array of foods in a variety of settings, reducing the likelihood that they will waste the food they choose to consume. Therefore, it is unlikely that similar levels of food waste would be observed outside a hospital setting. It is unclear whether or not a proportional difference in food waste between vegetarian and meat-containing meals would be maintained outside a hospital setting. However, it is well known that the environmental impacts associated with meat are greater than those associated with most vegetarian foods, so it is reasonable to expect that food waste from meat-containing meals would still have higher GHGE for a similar amount of food wasted.

## 5. Conclusions

It is important both to reduce the GHGE associated with food provision and reduce the proportion of food that goes to waste as part of efforts to limit the negative environmental consequences of food systems. Fortunately, the case study examined here provides an example where one choice—serving vegetarian meals to patients by default for their first 24 h in a hospital setting—improves both outcomes. Food waste from vegetarian meals was lower in both total weight and associated GHGE than food waste from meat-containing meals. 

## Figures and Tables

**Figure 1 nutrients-14-01174-f001:**
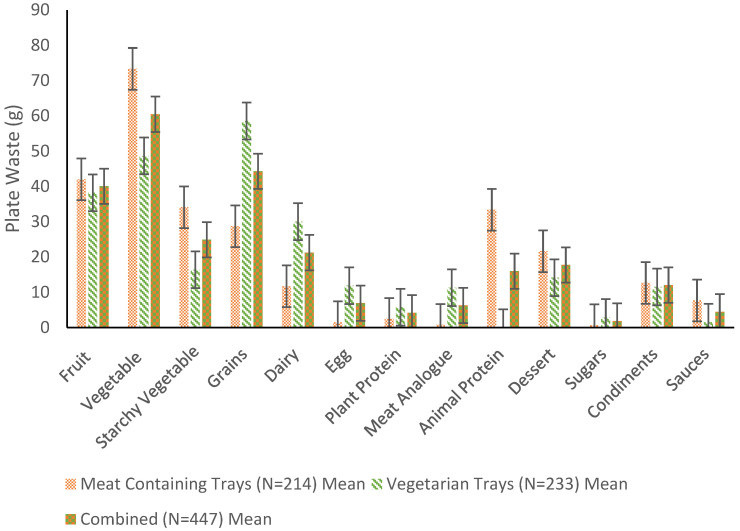
Categories of foods and their respective amounts of waste differentiated by meal type. Error bars represent the standard error of mean.

**Figure 2 nutrients-14-01174-f002:**
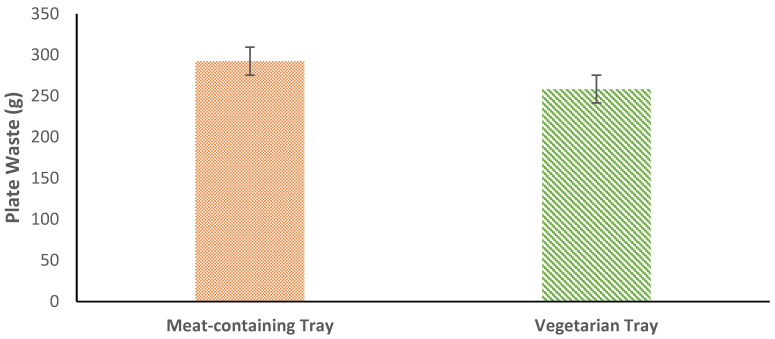
Plate waste from meat-containing and vegetarian meals. Vegetarian meals had less FW than meat-containing meals. Error bars represent the standard error of mean.

**Figure 3 nutrients-14-01174-f003:**
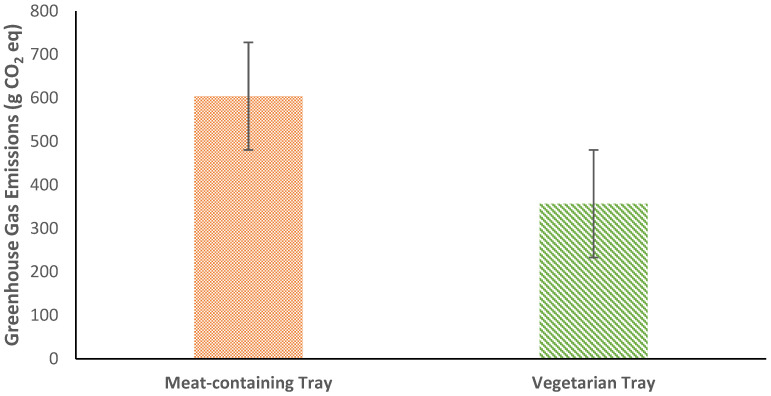
Mean total GHGE (g CO_2_ eq) by meal type. Error bars represent the standard error of mean.

**Table 1 nutrients-14-01174-t001:** Exploratory comparison analyses for food waste (g/plate) and GHGE (g CO_2_ eq/plate) between meat-containing and vegetarian meal types by food category.

Food Waste (g/plate)	Total						
	(N = 447)						
	M	±	SD	SE	*t*-Statistic	*p*-Value	Hedges’ g
Fruit	40.03	±	50.19	2.37	0.80	0.43	0.08
Vegetable ^1^	140.06	±	65.51	3.10	4.60	<0.001	0.44
Grains	44.27	±	54.73	2.59	6.10	<0.001	0.57
Dairy	21.26	±	45.69	2.16	4.40	<0.001	0.41
Egg	6.90	±	17.79	0.84	6.62	<0.001	0.61
Plant Protein ^2^	4.20	±	17.96	0.85	2.03	0.049	0.19
Dessert	17.74	±	35.54	1.68	2.22	0.03	0.21
**GHGE (g CO_2_ eq/plate)**							
Fruit	19.56	±	29.32	1.39	0.90	0.38	0.09
Vegetable ^1^	30.27	±	33.16	1.57	4.17	<0.001	0.39
Grains	41.61	±	54.44	2.57	4.38	<0.001	0.41
Dairy	77.84	±	161.04	7.62	3.82	<0.001	0.36
Egg	23.40	±	60.18	2.85	6.14	<0.001	0.61
Plant Protein ^2^	5.32	±	19.93	0.94	2.42	0.008	0.23
Dessert	108.08	±	323.83	15.32	2.67	0.004	0.25

^1^ Includes vegetables and starchy vegetables; ^2^ Plant protein items consist of peanut butter, tofu, black beans, brown lentils, and hummus.

## Data Availability

Data supporting results are available in Appendix A.

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
