# Peer review of "Comparison of Plate Waste between Vegetarian and Meat-Containing Meals in a Hospital Setting: Environmental and Nutritional Considerations"

_nutrients, 2022, doi:10.3390/nu14061174_

Round 1

Reviewer 1 Report

Thank you for this occasion to contriubte in helping to improve your intesting paper, congrats to this nice 1st version!

From the beginning on until the end, I loved and enjoyed reading this paper;
it is small and nice and has a strong message to deliver - if it will be elaborated focued (has to be done!), and if it will be better "sold" to an intersted reader - and i am sure the authors can deliver the core message better
i am looking FWD to the revised paper, congrats to this first version already

Major point that goes through the whole paper (mixing up content form Intro-Method-Results-Discussion) = improving academic writing (guess students/PhD candidates on their way to thesis :-))
2nd major point is to focus more on design of grafics n figures so that they are more conscise and informative; and also formatting the manuscript 
togehter both: it the current form not adequate for an scientific journal of nearly SIF=6 points -> but I am convinced after a focused revision :-)

Pls see my comments in the PDF file - all good success, of that i am convinced!

Abstract - compact Intro, sell better the conclusion/practical implication

Intro - mostly minor comments to improve, except of the last paragraph that is a bit mixing a bit withg method (i feel, not clear) but clarly lacking the straight line from identifying the gap in lit -> point out the need/relevance for general/global pop./etc. for some target group -> therefore, the AIM ...has to be fixed by improved academic writing

Method - some open Qs to answer

Results - mix up with method and discussion, not sober as it must be

Discussion - again mix with other chapters; too short - see my comment at the beginning, especially can be extended on basis of additional stats calucalations (see my comments before in paper) that will add meaning to the already intersting results/paper; all is there but not sufficiently put into comparision, current body of science etc.

Limiation - missing

Conclusion - not complete/sufficient

Reviewer 2 Report

This paper addresses the important question whether food waste is higher with vegetarian dishes compared to those containing meat. This comparative study was conducted in a hospital setting where patients automatically received a vegetarian diet on the first day upon admission and were supplemented with a meat-containing one on the consecutive days only if they requested it. The authors found that, compared to the meat-containing dishes, the leftovers from the vegetarian meals were lower, further saving greenhouse gas emissions on top of the already lower GWP of vegetarian meals.

However, the authors ignore existing statistical textbook knowledge on how to process non-normally distributed data and need to re-evaluate the data using either transformed data or applying non-parametrical tests. This is of particular relevance as this refers to their main outcome data. Moreover, the presentation of the figures can be improved.

Abstract

I perceive the Abstract as being too long given the information it contains. Everything written beyond Line 24 (starting with “Plate FW …”) is fine, but the text before that part can easily be shortened by one third.

Lines 59-70: All what has been written in this paragraph holds true, but one should also critically mention in this context that, compared to meat, some trace elements have a lower bioavailability if deriving from plant-based foods (i.e. iron) and that vegans and (partly) vegetarians have a higher risk of developing nutrient deficiencies, for e.g. iron, vitamin D, calcium, and vitamin B12 as compared to omnivores (unless the vegetarian food is fortified).

Line 67 Replace “beans categories” with “pulses”.

Lines 84-87: There is some redundancy in the information provided here and the description of the procedure under “Materials and Methods” – please remove.

Materials and Methods

Lines 123-124: The authors did not describe what they did if the data were not normally distributed (which they were not, due to Lines 147-150). Text-book consequences of processing non-normally distributed data would be transformation of data prior to testing (square root or log) or the application

Line 124-125: What were the criteria to consider a value to be an outlier?

Results

Line 133: How could have been more data analyzed from trays than had been collected?

Line 137-139: This statement should be reformulated for better informative content. How did the food waste and GWP differ across major food categories? Were they higher or lower with the vegetarian meals? Or did it differ by category whether there was more waste from the meat-containing dishes? If this sentence refers to Table 1, then avoid redundancy and remove this statement.

Lines 147-149 and 192-193. Unclear why the authors tested for ND at all if they anyway decide to ignore prerequisite test results for parametrical testing. I really do believe that proper evaluation demands either transforming the data before testing or using a non-parametric test (Mann-Whitney-U test).

Figures 1+2+3: I think that the graphs would look more appealing if standard error of mean (instead of SD) would have been used here.

Tables 2+4: If available, I would suggest including here also the average weight of meat waste – for orientation purposes.

Discussion

Line 234: If there are “a few” studies, please cite those and not only one.

I miss at least a short paragraph in the Discussion where the authors speculate on whether their results can be extrapolated to a more general setting (outside of a hospital).

Reviewer 3 Report

Although the topic of food waste is of scientific interest nowadays and offers many possibilities in its application, the authors, unfortunately, approached the topic in a very superficial way. The authors measured two parameters: Food Waste (FW) and Global Warming Potential (GWP). The main drawback of this paper is that the same data are presented in different tables and figures. 
For example, Appendix 1 and Figure 1 show the same data, while Table 1 is expanded by only three columns (t-statistics, p-value, and Hedge's g). Technically, all of this could be summarized in one table that is self-explanatory. 
The same is true for the GWP results - the same data are presented in Table 3, Figure 3, and Table 4. All these data can be summarized in one table. 
In addition, Table 2 is missing from the manuscript. 
In view of this, this manuscript contains interesting but insufficient amount of data and no new scientific findings. In my opinion, the quality of this manuscript is below the standards of your esteemed journal and I propose to reject it.

Round 2

Reviewer 1 Report

The manuscript has much improved, however, some minor but important points have still to be fixed before considered for publication - see my comments on some repeated issues. Good success!

Reviewer 2 Report

Besides very few exceptions, the authors have covered all points I raised during the first review of their manuscript.

First, the authors provided a compelling argumentation in their response for why they used parametrical tests despite the absence of normal distribution (Lines 172-174), so I can follow their line. Indeed, I would suggest that the explanation in the response to my comments should – in a compressed format –be included in the Material and Methods part ( or in Line 207) as it much more clearly points out the problem with switching over to non-parametrical testing. The text of the paper, as provided in the revised version, still leaves the reader puzzled with the question why ND testing was done at all if it didn’t have consequences anyway.

Second, in Line 10, please replace ” Vegetarian diets satisfy nutritional …” with “Vegetarian diets can satisfy nutritional …” as there are vegetarian diets that may not in every single case cover the recommended intake values.

In general, I would like to congratulate the authors on an important, very nicely and clearly written scientific article.

Reviewer 3 Report

The authors have significantly improved the paper.

Author Response

Response to Reviewer 3 Comments

Point 1: The authors have significantly improved the paper.

Response 1: We are glad to hear that, thank you.

This manuscript is a resubmission of an earlier submission. The following is a list of the peer review reports and author responses from that submission.